# FinEntity: Entity-level Sentiment Classification for Financial Texts [*]

**Yixuan Tang**[*], **Yi Yang** [*], **Allen H Huang** [*], **Andy Tam** [†], **Justin Z Tang** [†]

Hong Kong University of Science and Technology[*]

Hong Kong Monetary Authority [†]

{yixuantang,imyiyang,acahuang}@ust.hk

{akftam,jztang}@hkma.gov.hk

## Abstract

In the financial domain, conducting entity-level sentiment analysis is crucial for accurately assessing the sentiment directed toward a specific financial entity. To our knowledge, no publicly available dataset currently exists for this purpose. In this work, we introduce an entity-level sentiment classification dataset, called **FinEntity**, that annotates financial entity spans and their sentiment (positive, neutral, and negative) in financial news. We document the dataset construction process in the paper. Additionally, we benchmark several pre-trained models (BERT, FinBERT, etc.) and ChatGPT on entity-level sentiment classification. In a case study, we demonstrate the practical utility of using FinEntity in monitoring cryptocurrency markets. The data and code of FinEntity is available at https://github.com/yixuantt/FinEntity.

## 1 Introduction

*"We see ChatGPT's prowess and traction with consumers as a near-term threat to Alphabet's multiple and a boost for Microsoft and Nvidia."*[1] In this Wall Street Journal article, multiple financial entities are mentioned, but their sentiments are contrasting (Positive for Microsoft and Nvidia , and Negative for Alphabet ). In fact, a considerable portion of real-world financial text (such as news articles, analyst reports, and social media data) contains multiple entities with varying sentiment (Malo et al., 2014; Huang et al., 2023; Sinha and Khandait, 2021; Shah et al., 2023a). Nevertheless, most existing sentiment classification corpora in the financial domain are sequence-level, i.e., the sentiment label is associated with the entire text sequence. Consequently, these sequence-level sentiment datasets are unsuitable for the entity-level sentiment classification task.

Developing a natural language processing (NLP) system for entity-level sentiment classification necessitates the availability of a dataset with entity tagging and sentiment annotation. To our knowledge, no such public dataset currently exists. In this paper, we fill this gap by *constructing a new dataset that annotates both the financial entity spans and their associated sentiments within a text sequence*. We outline the development of a high-quality entity-level sentiment classification dataset for the financial domain, called **FinEntity**.

Subsequently, we benchmark several pre-trained language models (PLMs) and a zero-shot Chat-GPT model (gpt-3.5-turbo) on entity-level sentiment classification tasks. The results demonstrate that fine-tuning PLMs on FinEntity outperforms the zero-shot GPT model. This finding suggests that manually collecting a high-quality domain-specific dataset and fine-tuning PLMs is more suitable than relying on the zero-shot GPT model.

We further demonstrate the practical utility of FinEntity in investment and regulatory applications, extending the work of Ashwin et al. (2021); Wong et al. (2022). Collaborating with a regulatory agency, we apply the fine-tuned PLMs to a unique cryptocurrency news dataset. Experimental results indicate that the individual cryptocurrency sentiment, inferred using the FinEntity fine-tuned PLM, exhibits a stronger correlation with cryptocurrency prices than traditional sequence-level sentiment classification models. Furthermore, the inferred individual cryptocurrency sentiment can better forecast future cryptocurrency prices - leading to enhanced risk monitoring for regulatory agencies and investors.

We make the FinEntity dataset publicly available and hope it will be a valuable resource for financial researchers and practitioners in developing more accurate financial sentiment analysis systems.

---

[*] The views and analysis expressed in this paper are those of the authors and do not necessarily represent the views of HKMA. All errors are the authors' own.

[1]https://www.wsj.com/articles/microsoft-and-google-will-both-have-to-bear-ais-costs-11674006102

## 2 Related Work

**Financial Sentiment Classification.** NLP techniques have gained widespread adoption in the finance domain (Huang et al., 2023; Yang et al., 2023). One of the essential applications is financial sentiment classification (Kazemian et al., 2016; Yang et al., 2022; Frankel et al., 2022; Chuang and Yang, 2022). However, prior literature on financial sentiment classification focuses on the entire text sequence (Kazemian et al., 2016; Yang et al., 2022; Frankel et al., 2022). If a text paragraph contains multiple financial entities with opposing sentiment (as common in financial news or analyst reports), sentiment analysis for the entire text sequence may no longer be accurate. Consequently, a more fine-grained sentiment analysis approach is needed, one that is specific to individual financial entities.

**Financial Entity Tagging and Sentiment Dataset.** Existing financial sentiment classification datasets, such as Financial Phrase Bank (Malo et al., 2014), SemEval-2017 (Cortis et al., 2017), AnalystTone Dataset (Huang et al., 2023), Headline News Dataset (Sinha and Khandait, 2021) and Trillion Dollar Words (Shah et al., 2023a), are based on entire text sequence (sentence or article). FiQA, an open challenge dataset [2], features aspect-level sentiment; however, it does not include entity annotations. SEntFiN (Sinha et al., 2022) is a dataset for financial entity analysis in short news headlines. However, this dataset uses a pre-defined entity list to match entities and does not have entity tagging, so it is still not applicable to recognize financial entities from text. Moreover, most of the headlines contain only one entity, which makes it close to a sequence-level sentiment dataset. For financial entity tagging dataset, FiNER (Shah et al., 2023b) and FNXL (Sharma et al., 2023) are created for financial entity recognition and numeral span tagging respectively, but both lacks sentiment annotation. Therefore, we aims to bridge this gap by constructing a high-quality, entity-level sentiment classification dataset, which not only label the financial entity spans in sentences, but also annotate their associated sentiments.

## 3 Dataset Construction

**Initial Dataset.** We obtain a financial news dataset from Refinitiv Reuters Database. In the prescreening step, we utilize a pre-trained Named Entity Recognition model [3] and a sequence-level sentiment classification model [4] to infer the number of ORG entities and sequence sentiment. Subsequently, we compile a dataset with a balanced distribution of positive/negative/neutral articles, ensuring that 80% of the sequences contain more than one entity. Following prescreening, we obtain a dataset comprising 4,000 financial news sequences[5].

**Label.** Entity-level sentiment classification is a sequence labeling task. As such, we employ the BILOU annotation scheme. Specifically, each token in an input sequence is assigned one of the BILOU labels, indicating the beginning, inside, last, outside, and unit of an entity span in the sequence. Additionally, each annotated BILU entity is tagged with a sentiment label (positive/neutral/negative), while the O entity does not receive a sentiment label. Consequently, each token in the input sequence is assigned to one of thirteen labels (BILU-positive/neutral/negative and one O label).

**Annotators.** A total of 12 annotators are recruited, all of whom are senior-year undergraduate students majoring in Finance or Business at an English-speaking university. Three annotators label the same example to ensure data quality and perform cross-check validation. Therefore, each annotator is assigned 1,000 examples. We employ the LightTag platform [6] for the annotation job. Annotators are instructed to tag all named entities in the sequence and the sentiment associated with each entity. Focusing on the financial domain, we limit the named entity to companies (such as Apple Inc.), organizations (such as The Fed), and asset classes (such as equity, REIT, and Crude Oil). Annotators are advised not to tag persons, locations, events, or other named entities. A screenshot of the annotation interface is shown in Appendix A.

**Annotation Consistency.** A total of 28,631 entities are annotated by the annotators, and we conduct cross-checks in order to ensure data quality. Initially, we employ the Jaccard similarity coefficient to measure entity-level consistency between pairs of annotators. The overall Jaccard similarity of the dataset is 0.754. Furthermore, the number of examples with a Jaccard similarity equal to 1.0 is 44.35%, indicating that 44.35% of examples

---

[2]https://sites.google.com/view/fiqa/home

[3]https://huggingface.co/dslim/bert-base-NER
[4]https://huggingface.co/yiyanghkust/finbert-tone
[5]A sequence consists of multiple sentences.
[6]https://www.lighttag.io/

|  | Positive | Negative | Neutral | Total |
|---|---|---|---|---|
| Number | 503 | 498 | 1,130 | 2,131 |
| Percentage | 23.60% | 23.37% | 53.03% | 100% |

Table 1: Sentiment Label Distribution of Entities

|  | Single Entity | Multiple Entity | Total |
|---|---|---|---|
| Number | 390 | 589 | 979 |
| Percentage | 39.83% | 60.16% | 100% |

Table 2: Single/Multiple Entity Distribution

|  | BERT | BERT-CRF | FinBERT | FinBERT-CRF | ChatGPT (zero-shot) | ChatGPT (few-shot) |
|---|---|---|---|---|---|---|
| Negative | 0.75 | 0.82 | 0.83 | **0.88** | 0.58 | 0.62 |
| Postive | 0.81 | 0.81 | 0.81 | **0.84** | 0.39 | 0.73 |
| Neutral | 0.82 | 0.81 | **0.84** | 0.82 | 0.71 | 0.61 |
| Micro Avg | 0.80 | 0.81 | 0.83 | **0.84** | 0.59 | 0.67 |
| Macro Avg | 0.80 | 0.81 | 0.83 | **0.85** | 0.56 | 0.65 |
| Weighted Avg | 0.80 | 0.81 | 0.83 | **0.84** | 0.59 | 0.68 |

Table 3: Entity-level Sentiment Classification Results.

in the dataset have exactly the same [begin, end] span by all three annotators. We filter this subset for further sentiment consistency checks. Subsequently, we utilize Fleiss' Kappa to measure each example's sentiment annotation consistency (Gwet, 2014). We select examples with a Fleiss' Kappa higher than 0.8 and then use majority voting to obtain the entity's sentiment, ensuring high consistency in sentiment annotations.

**Final Dataset: FinEntity.** The final FinEntity dataset contains 979 example paragraphs featuring 503 entities classified as Positive, 498 entities classified as Negative, and 1,130 entities classified as Neutral, resulting in a total of 2,131 entities. Table 1 and Table 2 are detailed distrutions of FinEntity. The sentiment label distribution of entities is fairly balanced. Moreover, About 60% of the financial text in the dataset contains multiple entities. A sample of FinEntity is shown in Appendix B. Our ensuing analysis is based on the FinEntity dataset.

## 4 Benchmarking Entity-level Sentiment Classification

**PLMs.** We benchmark several PLMs for entity-level sentiment classification. We consider fine-tuning BERT (bert-base-cased) (Devlin et al., 2018) and a finance-domain specific FinBERT (Yang et al., 2020) by incorporating a linear layer on top of each token's hidden output. In order to account for token label dependency, we replace the linear layer with a conditional random field (CRF) layer, yielding BERT-CRF and FinBERT-CRF respectively. We implement those PLMs using the `transformers` library (Wolf et al., 2019).

**ChatGPT.** For comparison with state-of-the-art generative LLMs, we examine the zero-shot and few-shot in-context learning performance of ChatGPT by querying OpenAI's gpt-3.5-turbo model with a 0.0 temperature value. Following previous literature (Shah et al., 2023a), we construct a

prompt designed to elicit structured responses. The detailed prompt for zero-shot and few-shot learning is shown in Appendix C.

**Evaluation.** We randomly partition the dataset into 80% training dataset and 20% testing dataset. We employ Seqeval (Nakayama, 2018) for evaluation, reporting F1-scores, which include the negative, positive, and neutral classifications, respectively. It is important to note that for entity-level sentiment classification, a testing example is considered correctly classified only if all the named entities in the example are correctly tagged and their associated sentiments are correctly classified. This implies that this task is much more challenging than traditional sequence-level sentiment classification tasks.

**Results.** We present the benchmarking results in Table 3. These results demonstrate that fine-tuning PLMs exceeds the performance of ChatGPT model, in line with (Rogers et al., 2023) which suggests that zero-shot ChatGPT is not a strong baseline for many NLP tasks. The results provide important implications that for domain-specific, customized NLP tasks, manually collecting a high-quality dataset, through more labor-intensive, indeed achieves better performance than the current state-of-the-art generative LLM.

## 5 Case Study: Cryptocurrency Market

In this section, we demonstrate the practical utility of the FinEntity dataset for investment and regulatory applications. As a case study, we focus on the cryptocurrency market, which is notoriously volatile and has been plagued by instances of manipulation and fraud, as evidenced by the recent FTX crash. As such, it is crucial for regulators and investors to closely monitor the market sentiment toward different cryptocurrencies.

In this case study, we collaborate with a regulatory agency responsible for overseeing the monetary and financial systems in both local and global markets. The team at the regulatory agency shares a cryptocurrency news dataset (non-overlapping with our Reuters data) that they have internally col-

lected. The dataset comprises 15,290 articles spanning from May 20, 2022 to February 1, 2023. Each article is associated with a timestamp, enabling time-series analysis. We select four cryptocurrencies (Bitcoin, Ethereum, Dogecoin, Ripple) as the target entities for analysis, owing to their substantial market dominance and the attention they receive from investors.

## 5.1 Sentiment Classification

**Traditional approach: sequence-level.** To facilitate comparison, we utilize a sequence-level pre-trained financial sentiment classification model (Huang et al., 2023). For each focal cryptocurrency, such as Bitcoin, we extract sentences containing the target word (e.g., Bitcoin, BTC) from articles on the same date and feed them into the model to obtain sentiment labels (positive/negative/neutral). **Our approach: entity-level.** We employ the FinBERT-CRF model, which is fine-tuned on the FinEntity dataset, to extract daily entity-level sentiment. Specifically, we feed an article into FinBERT-CRF and obtain a set of named entities along with their associated sentiments. Subsequently, we group entities and aggregate sentiments to derive daily sentiment labels.

## 5.2 Contemporaneous Correlation Analysis

We begin by measuring the contemporaneous correlation between daily focal cryptocurrency prices and inferred sentiments. To obtain a numerical value for the sentiment score, we code positive labels as +1, neutral as 0, and negative as -1. Then, We calculate the sum of sentiment scores contained in each day's articles and normalize them using min-max normalization.

For illustration purposes, we present the correlation graph for Bitcoin in Figure 1. The graph indicates a positive correlation between the price and sentiment score inferred from the entity-level sentiment model FinBERT-CRF. Additionally, we observe that both the sentiment and the price of Bitcoin experienced a sharp decline in early November 2022, attributable to the bankruptcy of FTX.

Next, we compute the maximum information coefficient (MIC) between the cryptocurrency price and the inferred sentiment. Figure 2 displays a moderate positive correlation. Furthermore, entity-level sentiment exhibits higher correlations than sequence-level sentiment. Cryptocurrency markets are highly interconnected and frequently covered together in the press. Upon examining this dataset,

we find that 37.53% of the examples contain more than one entity. Among these examples, 12.50% include entities that have opposing sentiments. Thus, the entity-level sentiment model can more accurately infer the sentiment of a focal cryptocurrency compared to traditional sequence-level models.

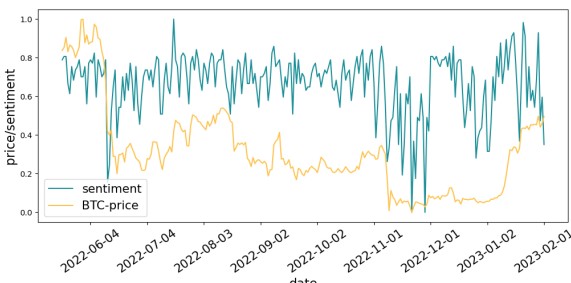

Figure 1: Daily Bitcoin prices and the sentiments inferred from FinBERT-CRF model.

## 5.3 Prediction Experiment

We also conduct a forecasting task where we predict the next day's Bitcoin price using its price time series and the inferred sentiment. For the price feature, we utilize the Open-High-Low-Close (OHLC) price. For sentiment feature, we opt for using the percentage of three different sentiment labels for each day as features. We chronologically divide the dataset into three parts: 60% for training, 20% for validation, and 20% for testing. We employ an LSTM as the prediction model. It incorporates a time step of 10 and a hidden size of 128.

We consider LSTM model that uses the OHLC price only and LSTM models that incorporate additional sentiment features inferred from either sequence-level or entity-level models. Table 4 reveals that the model incorporating entity-level sentiment features exhibits better accuracy than those utilizing sequence-level sentiment features or excluding sentiment features altogether. This further emphasizes that sequence-level approach is insufficient due to the presence of multiple entities within

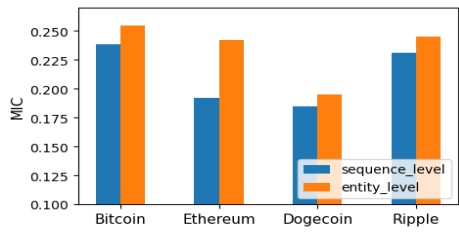

Figure 2: Correlation between sequence-level and entity-level sentiment and different cryptocurrency prices.

| Features | RMSE |
|---|---|
| OHLC + entity-level sentiments | 0.08502 |
| OHLC + sequence-level sentiments | 0.09549 |
| OHLC only | 0.11218 |

Table 4: Bitcoin price prediction performance.

a single sequence, highlighting the practical utility of our manually curated FinEntity dataset.

## 6 Conclusion

In this paper, we present the construction of FinEntity, a dataset with financial entity tagging and sentiment annotation. We benchmark the performance of PLMs and ChatGPT on entity-level sentiment classification and showcase the practical utility of FinEntity in a case study related to the cryptocurrency market. We believe that FinEntity can be a valuable resource for financial sentiment analysis in investment and regulatory applications.

## Limitations

First, our dataset construction relies on Reuters news. We have not extensively investigated the transferability of this news dataset to other financial corpora, such as corporate reports or social media posts. Second, since the dataset is in English, its applicability to non-English financial texts may be limited. Besides, financial decisions should consider multiple factors since relying on the above-mentioned forecasting methods may entail risks.

## Ethics Statement

The study was conducted in accordance with the ACL Ethics Policy. The annotators were recruited via a University Research program. All activities, including data annotation, comply with the University and program policies. Confidentiality and anonymity were ensured throughout the study by using unique participant identifiers and securely storing the data.

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

## A   Appendix A: Annotation Interface

A screenshot of the annotation interface is shown in Figure 3. Annotators tag one of the three sentiment labels (Positive, Negative, Neutral) to the selected named entities.

## B   Appendix B: A sample of FinEntity

We present a sample of FinEntity and its annotation in Table 5.

## C   Appendix C: Zero-shot and few-shot ChatGPT

We follow prior literature (Shah et al., 2023a) to construct the following prompt to query ChatGPT (OpenAI's gpt-3.5-turbo API) to obtain structured responses. Given a text sequence, ChatGPT returns a list of financial named entities (in the format of [begin, end] spans) and their associated sentiments.

*"Discard all the previous instructions. Behave like you are an expert entity recognizer and sentiment classifier. Identify the entities which are companies or organizations from the following content and classify the sentiment of the corresponding entities into 'Neutral' 'Positive' or 'Negative' classes. Considering every paragraph as a String in Python, provide the entities with the start and end index to mark the boundaries of it including spaces and punctuation using zero-based indexing. In the output, Tag means sentiment; value means entity name. If no entity is found in the paragraph, the response should be empty. The paragraph:{paragraph}.".*

For few-shot in-context learning, we concatenate three examples, i.e., three-shot, to the above prompt, as follows: *"user: "Other U.S. companies have made similar moves, including social media site Reddit Inc and Mobileye, the self-driving car unit of Intel Corp <INTC.O>. " assistant: {"start": 74, "end": 84, "value": "Reddit Inc", "tag": "Neutral"},{"start": 128, "end": 138, "value": "Intel Corp", "tag": "Neutral"}user: "Kellogg <K.N>, however, based the corporate headquarters for its largest business, snacks, in Chicago after announcing a split into three independent companies this summer. [nL4N2Y822D] "assistant: "{"start": 4, "end": 11, "value": "Kellogg", "tag": "Neutral"} "user: "Rival Oracle <ORCL.N> says in a statement on its website it has withdrawn all products, services and support for Russian and Belarusian companies, subsidiaries and partners. An Oracle spokesperson declined further comment."assistant:{"start": 183, "end": 177, "value": "Oracle", "tag": "Neutral"},{"start": 6, "end": 12, "value": "Oracle", "tag": "Neutral"}".*

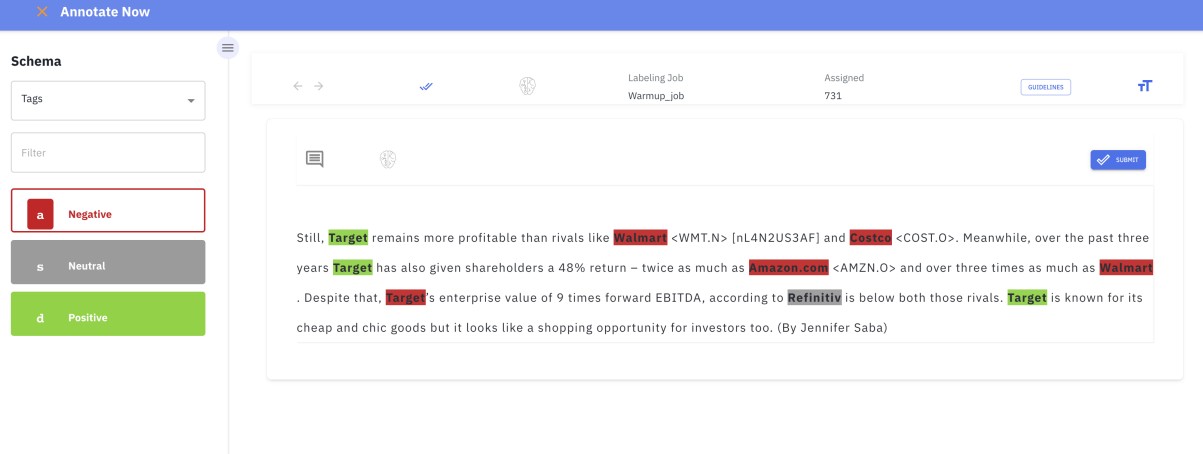

Figure 3: Entity-level Sentiment Annotation Interface.

| |
|---|
| "With S&P 500 earnings expected to grow 8.4% in 2022, the backdrop for stocks appears to be a solid one. However, skittish investors have punished companies such as Netflix <NFLX.O>, JPMorgan <JPM.N>and Tesla <TSLA.O>delivering less than stellar news in recent weeks, adding to the uneasy mood. Another large batch of reports is due next week, including from heavyweights Alphabet <GOOGL.O>and Amazon <AMZN.O>. " |
| {"start": 5","end": 12, "value": "S&P 500", "tag": "Positive"}, {"start": 164,"end": 171, "value": "Netflix", "tag": "Negative"} |
| {"start": 182,"end": 190, "value":"JPMorgan","tag": "Negative"}, {"start": 203,"end": 208, "value": "Tesla","tag": "Negative"} |
| {"start": 373","end": 381, "value": "Alphabet", "tag": "Neutral"}, {"start": 396,"end": 402, "value": "Amazon", "tag": "Neutral"} |

Table 5: Top row: A sample of FinEntity. Bottom row: Its entity span tagging and sentiment annotations.