# OpenReview forum: "FinEntity: Entity-level Sentiment Classification for Financial Texts"
_EMNLP/2023/Conference — EMNLP 2023 Main_

### Official Review · Reviewer_kPZc · 2023-08-03

**Typos Grammar Style And Presentation Improvements:** 1. line 062, "Experiment results" sho…
**Soundness:** 3

**Excitement:**

3: Ambivalent: It has merits (e.g., it reports state-of-the-art results, the idea is nice), but there are key weaknesses (e.g., it describes incremental work), and it can significantly benefit from another round of revision. However, I won't object to accepting it if my co-reviewers champion it.

**Paper Topic And Main Contributions:**

This paper focuses on constructing the FinEntity dataset for the financial domain, involving financial entity labeling and sentiment annotation. The authors also present the benchmarks for PLMs and ChatGPT in terms of entity-level sentiment classification. Experimental results validate the practice of the proposed dataset.

**Questions For The Authors:**

1. Regarding the FinEntity dataset, it is recommended that the detailed distributions, i.e., entity categories, single entity, and multiple entity distributions, be presented in tabular form.
2. As claimed in the limitations, only 4000 samples are annotated, and more domains, i.e., corporate reports or social media posts can be added.

**Reasons To Accept:**

1. The annotated dataset may be useful to the community;
2. The authors describe the detailed process of structuring the FinEntity dataset.


**Reasons To Reject:**

1. The detailed distribution of the proposed dataset is unclear;
2. Only three entities (company, organization, asset class) are annotated;
3. The experiments are a bit simple.

**Reproducibility:**

3: Could reproduce the results with some difficulty. The settings of parameters are underspecified or subjectively determined; the training/evaluation data are not widely available.

**Reviewer Confidence:**

4: Quite sure. I tried to check the important points carefully. It's unlikely, though conceivable, that I missed something that should affect my ratings.

---

> ### Author Rebuttal · Authors · 2023-08-27
>
> Thank you for your insightful comments. We are happy that you find our work "may be useful to the community." Yes, we will release the dataset.
>
> Our response to your comments is as follows.
>
> 1. Your suggestion on adding detailed data distribution is well taken. The distribution is shown below. As you can see, about 75% of financial text in the dataset contains multiple entities. Moreover, the sentiment label distribution of entities is fairly balanced.
>
> | |Positive | Negative | Neutral | Total |
> |-------------|-----------|----------|---------|-------|
> | Number     | 1,586     | 1,261    | 3,041    | 5,888  |
> | Percentage | 26.94%| 21.42% | 51.64%  | 100%  |
>
>
> |                    | Single Entity | Multiple Entity | Total |
> |--------------|--------------- |-----------------|-------|
> | Number      | 457              | 1,438               | 1,895  |
> | Percentage |  24.11%      | 75.88%          | 100%  |
>
> 2. Regarding your comment that "there are only three entities are annotated," we apologize for the confusion caused. The FinEntity dataset contains 1,921 unique entities (such as Tesla Inc., AAPL, S&P 500) belonging to three types (company, organization, and asset class). We focus on these three types because they are the most prevalent in financial text. Our annotation pipeline allows us to include more financial entity types in future extensions.
>
> 3. We agree with your comment that "corporate reports or social media posts can be added." We have already set up the annotation interface so that we can easily extend the dataset in the future. We will discuss this future direction in the revised manuscript.
>
> We thank you for your insightful comments. Should you have any further comments, we are happy to address them.

---

### Official Review · Reviewer_NjN2 · 2023-08-04

**Typos Grammar Style And Presentation Improvements:** none
**Soundness:** 3

**Excitement:**

2: Mediocre: This paper makes marginal contributions (vs non-contemporaneous work), so I would rather not see it in the conference.

**Missing References:**

Du, Chi-Han, Ming-Feng Tsai, and Chuan-Ju Wang. "Beyond word-level to sentence-level sentiment analysis for financial reports." ICASSP 2019-2019 IEEE International Conference on Acoustics, Speech and Signal Processing (ICASSP). IEEE, 2019.

Lin, Sheng-Chieh, et al. "Self-attentive sentimental sentence embedding for sentiment analysis." ICASSP 2020-2020 IEEE International Conference on Acoustics, Speech and Signal Processing (ICASSP). IEEE, 2020.

**Paper Topic And Main Contributions:**

The paper addresses the disparity in sentiment classification within the financial domain by introducing the FinEntity dataset. This dataset annotates financial entities and their corresponding sentiments in financial news, facilitating entity-level sentiment classification. Pre-trained language models fine-tuned with FinEntity outperform zero-/few-shot ChatGPT models in this task. The practical utility of the dataset is demonstrated through a case study on the cryptocurrency market, showcasing stronger correlations between cryptocurrency prices and inferred sentiment. An LSTM model incorporating entity-level sentiment features shows improved accuracy in Bitcoin price prediction. Overall, the FinEntity dataset offers a valuable resource for accurate financial sentiment analysis systems.

**Questions For The Authors:**

In section 5.1, the entity-level approach used in the paper is not explicitly clarified regarding the grouping of entities. It is unclear whether entities are grouped based on the target words utilized in the sequence-level approach or if the grouping and aggregation of sentiments are done manually. Additional information on the entity-level approach's methodology would help to understand the process better and ensure the validity of the comparison with the sequence-level approach.

**Reasons To Accept:**

1. Clear Documentation: The paper provides transparent and detailed documentation of the data collection and sentiment annotation process, allowing easy dataset reproduction and potential expansion by other researchers.
2. Objective Sentiment Annotation: The method used to assign sentiment labels to each financial entity appears to be objective, ensuring consistency in the dataset construction process.

**Reasons To Reject:**

1. Lack of Innovation: The dataset construction method employed in the paper is relatively simple and lacks innovation, leading to concerns about the originality and novelty of the research.
2. Experimental Report Style: The paper's approach and presentation make it resemble more of an experimental report rather than a research paper, potentially undermining its overall contribution to the field.
3. Ambiguity in Dataset Size: In section 5.1, it is not explicitly clear whether the dataset size used for sequence-level and entity-level models is the same. The distinction between the two models in terms of data utilization needs clarification to ensure a fair comparison.

**Reproducibility:**

4: Could mostly reproduce the results, but there may be some variation because of sample variance or minor variations in their interpretation of the protocol or method.

**Reviewer Confidence:**

4: Quite sure. I tried to check the important points carefully. It's unlikely, though conceivable, that I missed something that should affect my ratings.

---

> ### Author Rebuttal · Authors · 2023-08-27
>
> Thank you for your insightful comments. Our response to your comments is as follows.
>
> 1. We respectfully disagree with your comment that "the dataset lacks innovation." The primary contribution of this work is a new dataset for financial domain entity-level sentiment analysis. To our knowledge, this is the first dataset of such kind. We do not present a new method for the task. Moreover, the recent NLP community has made several notes on how to evaluate the contribution of a Resource paper [1,2]. We would argue that FinEntity is a high-impact, high-quality data set (see [1, 2]), which makes it an appropriate contribution to the Resources and Evaluation Track.
>
> 2. Regarding your "Experimental Report Style" comment, we would like to refer to Prof. Anna Roger's note (Myth 1: Resource papers are not science) [2], and we believe that our construction of FinEntity, an entity-level sentiment annotation for the financial domain, makes a clear contribution to the field. Recent literature has also noted the importance and the need to evaluate financial NLP tasks using financial-domain specific datasets (Shah et al. 2022).
>
> 3. Regarding your question on the "dataset size in Section 5.1", we apologize for any confusion caused. Both sequence-level and entity-level models use the same dataset, composed of sentences from financial news containing focal cryptocurrency entities (e.g., Bitcoin, BTC). This results in an identical data size. The distinction between sequence-level and entity-level models lies in the level of sentiment analysis: entity-level analysis uses the entity-level sentiment as the focal cryptocurrency sentiment, whereas sequence-level analysis uses the entire sequence sentiment as its sentiment. Our experiment shows that entity-level sentiment analysis can better capture the focal cryptocurrency's sentiment than the conventional sequence-level analysis. We will clarify it in the revised manuscript.
>
> We thank you for your insightful comments. Should you have any further comments, we are happy to address them.
>
> [1]: Advice on Reviewing for EMNLP, EMNLP 2020 Blog post, May 17th, 2020
> [2]: Peer review in NLP: resource papers, Anna Rogers, April 16th, 2020

---

### Official Review · Reviewer_R2TV · 2023-08-04

**Soundness:** 4

**Ethical Concerns:**

Yes

**Excitement:**

4: Strong: This paper deepens the understanding of some phenomenon or lowers the barriers to an existing research direction.

**Justification For Ethical Concerns:**

Authors did not provide a dedicated "Ethics Statement". My main concern is related to the use of 12 undergraduate students for labeling their dataset. It is not clear if this comply with good scientific ethics standards.

**Paper Topic And Main Contributions:**

In this work, the authors introduce a new entity-level sentiment analysis classification dataset focused on Financial Texts. Entity-level sentiment analysis (SA) is a branch of SA where sentiment is associated with specific entities in the text and is not just an overall sentiment classification common in sentence-level sentiment analysis. The authors describe step-by-step how they collected their data and annotated their dataset. Moreover, they propose a specific classification model (FinBERT-CRF) and show that it outperforms other transformed-based models. In a case example, the authors also demonstrate how one may use their dataset to build a model that predicts cryptocurrency prices better than without using the model.

**Reasons To Accept:**

- The dataset created by the authors is interesting, and it is a clear contribution to the community when publicly available, especially since the entity-level dataset is much more scarce than the sentence-level.
- The authors demonstrate care and attention to quality when creating their dataset. Also, they demonstrate a pertinent application of their dataset in a real-world scenario for stock prices forecast.
- The paper is quite direct and easy to follow.

**Reasons To Reject:**

- There is no Ethical Authors did not provide a dedicated "Ethics Statement". My main concern is related to the use of 12 undergraduate students for labeling their dataset. It is not clear if this comply with good scientific ethics standards.
- The authors does not carefully state the potential problems and even financial risks one may have if they try do implement and use their approach to forecast currency prices.
- There are overstatements such "no such public dataset currently exists". There are many other entity-level datasets, including from SemEval-2022. Authors should clarify who their dataset are unique.

**Reproducibility:**

4: Could mostly reproduce the results, but there may be some variation because of sample variance or minor variations in their interpretation of the protocol or method.

**Reviewer Confidence:**

4: Quite sure. I tried to check the important points carefully. It's unlikely, though conceivable, that I missed something that should affect my ratings.

**Typos Grammar Style And Presentation Improvements:**

- Provide references of using conditional random field layer with BERT.
- In table 1 caption, state which metric the numbers mean, is it accuracy? F1 Score?

---

> ### Author Rebuttal · Authors · 2023-08-27
>
> Thank you for your insightful comments. We are happy that you find our work "is a clear contribution to the community when publicly available." Yes, we will release the dataset.
>
> Our response to your comments is as follows.
>
> 1. Regarding your "ethics concern," we want to assure you that the undergraduate students were recruited via a University Research program. All the activities, including the data annotation, comply with the University and program policy. To our knowledge, there is no ethical concern on this matter. We will add an "Ethics Statement" to the manuscript to alleviate your concern.
>
> 2. Regarding your comment that "potential problems and even financial risks one may have if they try to implement and use their approach to forecast cryptocurrency prices," please note that the primary purpose of constructing such a dataset is that we want to contribute a fine-grained and high-quality dataset to the financial NLP community so that financial sentiment analysis can be conducted in a more robust way. By contributing such a dataset, we hope to minimize potential unexpected financial risk caused by inaccurate sentiment analysis. However, we do agree with your comment, and we will add a discussion to the manuscript.
>
> 3. Regarding your comment that "there are many other entity-level datasets, including SemEval-2022", we agree that they are many entity-level datasets. However, to our best knowledge, many of them, including SemEval-2022, are not financial domain-specific. Recent literature has noted the importance and the need to evaluate financial NLP tasks using financial-domain specific datasets (Shah et al. 2022). As we discussed in the literature review section, there are a few entity-level datasets in the financial domain. However, none of them provides an entity-level sentiment label. Therefore, we believe FinEntity will be a high-impact, high-quality data set.
>
> Shah, Raj, et al. "When FLUE Meets FLANG: Benchmarks and Large Pretrained Language Model for Financial Domain." Proceedings of the 2022 Conference on Empirical Methods in Natural Language Processing. 2022.

---

### Meta-Review · Area_Chair_mWNm · 2023-09-21

**Recommendation:** 4

**Metareview:**

This study introduces an entity-level sentiment analysis dataset tailored for the financial domain. Alongside the dataset, the authors present baseline experiments utilizing various models, including BERT, FinBERT, and ChatGPT.

Reviewer R2TV appreciated the dataset and its development process. However, reviewer raised valid concerns about the ethics statement and potential implications of the model. These concerns can be addressed in the limitations section, emphasizing that the direct application of the models' decisions could have adverse effects. The authors have already indicated in their rebuttal that an ethics statement will be incorporated, which should be added.

Reviewer NjN2 appreciated the detailed description of the data collection and annotation process. To further enhance the paper's readability, it is advised to elaborate on the concerns regarding data size and distribution.

---

### Decision · Program_Chairs · 2023-10-07

**Decision:**

Accept-Main

**Comment:**

This study introduces an entity-level sentiment analysis dataset tailored for the financial domain. Alongside the dataset, the authors present baseline experiments utilizing various models, including BERT, FinBERT, and ChatGPT.

Reviewer R2TV appreciated the dataset and its development process. However, reviewer raised valid concerns about the ethics statement and potential implications of the model. These concerns can be addressed in the limitations section, emphasizing that the direct application of the models' decisions could have adverse effects. The authors have already indicated in their rebuttal that an ethics statement will be incorporated, which should be added.

Reviewer NjN2 appreciated the detailed description of the data collection and annotation process. To further enhance the paper's readability, it is advised to elaborate on the concerns regarding data size and distribution.